# PROSTest, a Novel Liquid Biopsy Molecular Assay, Accurately Guides Prostate Cancer Biopsy Decision-Making in Men with Elevated PSA Irrespective of DRE Findings

**DOI:** 10.3390/cancers17243908

**Published:** 2025-12-06

**Authors:** Craig G. Rogers, Srinivas V. Koduru, Anthony Gulati, Abdel B. Halim

**Affiliations:** 1Vattikuti Urology Institute, Henry Ford Health, Detroit, MI 48202, USA; crogers2@hfhs.org; 2Wren Laboratories, Branford, CT 06405, USA; ahalim@wrenlaboratories.com; 3Bennett Cancer Center, Stamford, CT 06902, USA; agulati@stamhealth.org

**Keywords:** biopsy, DRE, prostate cancer, PROSTest, PSA, stratification

## Abstract

This study identified that PROSTest demonstrated high diagnostic accuracy (>90%) and reproducibility in risk stratification for PCa among men with PSA ≥ 3 ng/mL, independent of DRE findings or PSA levels. PROSTest provides robust and independent risk stratification across PSA levels and DRE and could minimize falsely alerted biopsies.

## 1. Introduction

Prostate cancer (PCa) is the second most commonly diagnosed cancer in men, with an estimated 1.4 million new cases and 375,000 deaths worldwide in 2020 [1,2]. Prostate cancer is the leading male cancer diagnosis in the majority of countries and the primary cause of cancer death in one-quarter globally [3]. In the US, 255,395 new PCa cases were reported in 2022, and 33,881 men died from the disease in 2023 [4].

The diagnostic pathway for PCa aims to ensure the timely detection of clinically significant PCa while avoiding invasive testing in indolent cases. Clinical symptoms, individual screening, and/or population-based screening all provide diagnostic pathway access. The prevalence of PCa, and particularly clinically significant PCa, however, varies by indication, resulting in different diagnostic yields across these entry points.

Prostate-specific antigen (PSA) is a glycoprotein enzyme secreted by prostate epithelial cells, with only a small portion detectable in the blood stream. It is the primary biomarker used in the evaluation of suspected PCa, and its introduction as a serum test has transformed PCa diagnosis [5]. PSA is organ-specific but not cancer-specific, meaning it may also be elevated in benign prostatic hyperplasia (BPH), prostatitis, and other non-malignant conditions. There is currently no universally accepted threshold for defining an abnormal PSA level [6]. It is a continuous parameter, with higher levels and rapid rising, indicating greater likelihood of PCa. However, some men may harbor PCa even when serum PSA levels are low [7].

PSA is widely used in screening and early detection strategies, often in combination with digital rectal examination (DRE). Prior to the detection of PSA, DRE was used to identify palpable suspicious areas on the prostate and to estimate prostate volume. While DRE remains a cost-effective and accessible diagnostic tool, it is inherently subjective and operator-dependent. African American and Afro-Caribbean men undergo DRE less often than White men [8,9], which reflects a fear of screening, despite having a higher prostate cancer worry [8]. DREs have unique psychological demands for men. This relates to vulnerability, humiliation, and sexuality, all of which have significant negative impacts on utility [9]. In the context of primary detection, DRE has moderate to poor diagnostic performance. Furthermore, the interpretation of DRE findings varies significantly among clinicians [2,10], contributing to its limited reliability. Despite this, DRE continues to be often used alongside PSA in PCa detection strategies in clinical guidelines [3,11].

In approximately 18% of cases, PCa is detected based on a suspicious DRE alone, regardless of PSA level [12]. A suspicious DRE in subjects with PSA level ≤ 4 ng/mL has a positive predictive value (PPV) ranging from 5% to 30% [12]. In the European Randomized Study of Screening for Prostate Cancer (ERSPC) trial, an abnormal DRE in conjunction with an elevated PSA more than doubled the likelihood of a positive biopsy (48.6% vs. 22.4%) [13]. An abnormal DRE is an indication for multiparametric MRI or direct biopsy when there is suspicion of extracapsular disease (cT3-4) [13,14].

An abnormal DRE is associated with an increased risk of surveillance [15], and remains a predictor of advanced PCa (OR: 11.12 for cT3 and OR: 5.28 for cT4) [16]. Both clinical T staging and the current European Association of Urology risk group stratification depend on DRE.

PROSTest is a recent (2024), novel, blood-based, multigene expression assay that evaluates whole-blood mRNA profiles of 27 prostate cancer-associated genes using quantitative PCR (qPCR) and a proprietary machine learning algorithm [17]. The assay yields a continuous score ranging from 0 to 100, with a validated binary cutoff of ≥50, which categorizes subjects as either “positive” (high probability of cancer) or “negative” (low probability of cancer). In contrast to PSA and other serum- or urine-based biomarkers that infer cancer risk indirectly [18,19,20,21], PROSTest captures direct gene expression signals associated with PCa through analysis of whole-blood mRNA at the molecular level. Two recent studies have evaluated the clinical utility of the PROSTest, identifying good positive predictive values (PPV) of 91.6–98.1% and negative predictive values (NPV) of 66.3–95.6% when used as a diagnostic across PSA strata [22,23].

We conducted a retrospective cohort study with a prespecified statistical analysis plan to test the null hypothesis that DRE findings do not significantly influence the sensitivity, specificity, or overall stratification performance of PROSTest. The objective of this study was to assess whether PROSTest, as a blood-based molecular biomarker, could improve the accuracy of PCa diagnosis by providing more refined risk assessment beyond the traditional stratification methods of PSA and DRE. We report findings from a retrospective analysis of PROSTest results in men ≥45 years of age with PSA levels ≥ 3 ng/mL who underwent biopsy as part of a routine screening evaluation.

## 2. Materials and Methods

### 2.1. Study Design

The study was conducted in subjects attending urology outpatient clinics for PCa screening between January 2022 and June 2024 (Appendix A). Eligibility criteria were based on age and PSA thresholds consistent with established screening guidelines, with inclusion restricted to subjects aged ≥45 years and a PSA threshold of ≥3 ng/mL [24,25]. Blood samples were collected for both PSA and PROSTest measurements and then all subjects underwent DRE. Systematic prostate biopsy was performed in all cases. A precision-based sample size estimation was performed for the primary diagnostic accuracy metrics. Assuming a prostate cancer prevalence of approximately 40% in the evaluated population and anticipated PROSTest sensitivity and specificity of ~80% based on pilot data, a minimum of 270 subjects was required to estimate both sensitivity and specificity with a two-sided 95% confidence interval width of ≤10%, which is consistent with accepted standards for the validation of diagnostic tests. The final cohort of 327 subjects exceeded this requirement and therefore provided adequate precision for estimating test performance across PSA and DRE subgroups.

### 2.2. Consent

All participants provided written informed consent to participate in the study (WIRB #20191473—WCG Clinical). The study is registered on ClinicalTrials.gov: RegisterPROS—a Registry for Prostate Cancers (NCT06872619).

### 2.3. Subjects

Subjects were evaluated according to the local protocol for primary cancer detection. Most presented with a suspicious DRE or clinical symptoms suggestive of PCa. No patient had undergone prostate-targeted therapy or prostate surgery, or had taken oral 5α-reductase inhibitors before biopsy.

### 2.4. DRE

DRE was performed in all subjects by board-certified urologists at each institution using a standardized clinical technique, with the patient in the left-lateral decubitus position. Findings were classified as either normal or suspicious for carcinoma if the digital rectal examination of the prostate revealed features such as firmness, nodularity, asymmetry, induration, or loss of anatomical landmarks [26,27,28].

### 2.5. Blood Collection

Following informed consent, a single blood sample was collected from each participant for PROSTest analysis. Samples were drawn into Wren’s proprietary RNA stabilization buffer tubes, in accordance with the standardized sample collection protocol [17,22]. Specimens were de-identified, coded, and shipped under controlled conditions for analysis. Upon arrival, samples were stored at −80 °C until batch analysis at Wren Laboratories (CAP# 8640840). A second blood sample was collected simultaneously for concurrent standard-of-care PSA measurements, which was performed via enzyme-linked immunosorbent assay (ELISA) in the clinical laboratory, following institutional protocols. PSA values were not normalized for prostate volume or BMI.

### 2.6. PROSTest Measurements

The PROSTest assay employed a 2-step workflow consisting of RNA isolation followed by reverse transcriptase quantitative (RT)-qPCR [29]. mRNA transcripts were isolated from whole blood using the Mini Blood Kit (Qiagen, Valencia, CA, USA) and real-time qPCR was performed on 27 target genes (Appendix A) using pre-spotted PCR plates (Life Technologies, Carlsbad, CA, USA) [17]. Target transcript levels were normalized to three endogenous references genes: *ALG9*, *TOX4*, and *TPT1,* and quantified using the ΔΔC_t_ method [17]. These normalized gene expression results were fed into a locked machine learning algorithm and expressed as a continuous risk score ranging from 0 to 100. A score ≥ 50% was interpreted as indicating an elevated risk of disease [17].

### 2.7. Statistics

Descriptive statistics were used to characterize patient demographic and clinical characteristics. Quartile evaluation of PROSTest was undertaken to assess the association between the biomarker scores and a PCa diagnosis. Each quartile was compared to the median quartile. The latter was also compared to the clinical cut-off of ≥50.

Diagnostic metrics for DRE, and PROSTest, including sensitivity, specificity, PPV, NPV, and accuracy, were calculated using standard 2 × 2 contingency tables. Sub-analyses were undertaken evaluating the impact of ethnicity and prostate cancer family history on PPV, NPV, and the accuracy of the PROSTest.

Multiple logistic regression analysis was used to assess the independent contributions of age, DRE positivity (DRE+), a family history of PCa, PSA > 10 ng/mL, and PROSTest positivity (PROSTest+) to the likelihood of PCa diagnosis. Exact 95% confidence intervals (CIs) and Odd’s ratios (ORs) were calculated for key proportions of interest.

A standard continuous variable AUC ROC was generated for both PROSTest and PSA to evaluate further cut-offs. A 3-point AUC ROC was then generated to demonstrate the clinical performance of DRE and PROSTest to identify the utility of binary outputs for PCa detection (positive/negative) in the whole cohort as well as in the populations with PSA 3–10 ng/mL and PSA > 10 ng/mL.

All statistical analyses were performed using PRISM 9.4.0 for Windows (GraphPad Software (Version 10.6.1), La Jolla, CA, USA, www.graphpad.com (accessed on 12 November 2025)) and MedCalc Statistical Software version 20.109 (MedCalc Software bvba, Ostend, Belgium; http://www.medcalc.org; 2017). A two-sided *p*-value < 0.05 was considered statistically significant.

## 3. Results

### 3.1. Cohort Characteristics and Cancer Detection

A total of 327 subjects who met the inclusion criteria of age ≥ 45 years with PSA ≥ 3 ng/mL were included in this retrospective analysis. Demographics are included in Appendix A. The median age was 66 years (range: 46–88); 23 (7%) were >75 years. The ethnic breakdown included the following: 2 (0.6%) Mixed ethnicity, 124 (37.9%) African American, and 201 (61.5%) White subjects. Forty-five individuals (13.8%) had a family history of PCa. Of the 327 subjects, 215 subjects (65.7%) had PSA values between 3 and 10 ng/mL (Cohort 1), while 112 subjects (34.3%) had PSA levels > 10 ng/mL (Cohort 2). All subjects underwent DRE, blood sampling for PROSTest, and prostate biopsy. One hundred and twenty-nine (39.4%) were DRE-positive.

PCa was diagnosed in 131 of 327 subjects (40.1%) overall—69 of 215 (32.1%) in Cohort 1 and 62 of 112 (55.4%) in Cohort 2 (Figure 1). This included 2/126 (1.6%: Mixed and African American) and 129/201 (64.2%: Whites).

#### Diagnostic Performance of DRE

DRE demonstrated limited utility in stratifying PCa risk. In Cohort 1, the sensitivity and specificity of DRE were 42.0% and 62.3%, respectively, with an overall diagnostic accuracy of 55.8%. In Cohort 2, sensitivity was 64.4%, specificity was 50.8%, and overall diagnostic accuracy of 56.3%.

### 3.2. Diagnostic Performance of PROSTest

#### 3.2.1. Quartile Evaluation of PROSTest

We initially used a quartile approach to evaluate different PROSTest cut-offs in this population. For these analyses, we performed logistic regressions comparing the median PROSTest with each other quartile as well as the median quartile (Q3 + Q4 vs. Q1 + Q2) against the standard PROSTest cut-off of 50 (Table 1). The OR for the median PROSTest score was 136.4 (range: 47.38–392.89). PROSTest + ve (cut-off of 50) exhibited the highest OR (156.7) and z-statistic (10.68) consistent with the most accurate cut-off for PCa detection.

#### 3.2.2. Cohort 1: Subjects with PSA 3–10 ng/mL

Among individuals in this PSA range (PCa risk > 15%), PROSTest positivity was strongly correlated with biopsy-confirmed PCa, regardless of DRE findings. In the DRE−/PROSTest+ group (*n* = 49), 38 cancers were detected, corresponding to a sensitivity of 96.6% (95% CI: 82.2–99.9%) and a PPV of 93.3% (95% CI: 79.3–98.1%) (Table 2).

In contrast, the DRE+/PROSTest− group (*n* = 52) had only one cancer detected, despite abnormal DRE findings, yielding a PPV of 1.9%. This suggests that DRE positivity has minimal predictive value in the absence of a positive PROSTest result.

The DRE+/PROSTest+ group demonstrated similarly high sensitivity (96.6%) and specificity (92.7%), comparable to the DRE−/PROSTest+ group. These results indicate that PROSTest diagnostic performance is not influenced by DRE status.

In a sub-analysis, we evaluated the relationship between ethnicity (African American vs. White) and a family history of PCa on the PPV, NPV, and accuracy of the PROSTest in this PSA cohort (Appendix A).

In the DRE+ group, ethnicity had no impact on these metrics (PPV: >95%, NPV 100%, Accuracy: 90.7–97.6%). In the DRE− group, metrics were similar (PPV > 95%, NPV 95–100%, accuracy > 95%).

In those with a positive DRE and a family history of the disease, all metrics were 100%, versus 60–100% in the DRE-ve group. Metrics were similar irrespective of DRE status in those with no family history (sensitivity: 87% vs. 82%; specificity: 97% vs. 97% and accuracy: 93% vs. 92%).

#### 3.2.3. Cohort 2: Subjects with PSA > 10 ng/mL

In subjects with PSA levels greater than 10 ng/mL (PCa risk > 50%), a similar diagnostic pattern was observed. In the DRE−/PROSTest+ subgroup (*n* = 37), 31 PCAs were detected, yielding a PPV of 83.8% (95% CI: 71.3–91.5%) and NPV of 93.3% (95% CI: 78.4–98.2%). These results were comparable to those in the DRE+/PROSTest+ group (*n* = 30), which exhibited a PPV of 93.3% and NPV of 98.1%.

In contrast, the DRE+/PROSTest− subgroup (*n* = 15) had only confirmed cancer, corresponding to a PPV of just 6.7%, again underscoring the limited standalone value of a positive DRE. Among DRE−/PROSTest− individuals (*n* = 30), only two cancers were found, confirming the high NPV (93.3%) of a negative PROSTest result, regardless of DRE status.

In a sub-analysis, the impact of ethnicity (African American vs. White) and a family history of PCa was evaluated for PROSTest PPV, NPV, and accuracy in this PSA cohort (Appendix A).

In the DRE+ group, ethnicity had no impact on these metrics (PPV: >95%, NPV 88–100%, Accuracy: >95%). In the DRE− group, metrics were similar (PPV: 100%, NPV 78–100%, accuracy: 95%).

In those with a positive DRE and a family history of the disease, all metrics were 100% versus 60-100% in the DRE-ve group. Metrics were similar irrespective of DRE status in those with no family history (sensitivity: 93% vs. 85%; specificity: 92% vs. 92% and accuracy: 93% vs. 88%).


**Head-to-Head Comparison (PSA and PROSTest)**


A direct comparison of PSA and PROSTest performance across PSA strata ≥ 3 ng/mL revealed that PROSTest significantly outperformed PSA (Figure 2). The continuous ROC analysis confirmed the utility of the PROSTest across all PSA values as well as within each of the PSA cohorts evaluated. The AUCs for PROSTest in the whole cohort and the two populations with PSA of 3–10 ng/L and >10 ng/mL were 0.971–0.974, while these were 0.53–0.73 for PSA. Associated cut-offs, sensitivities, specificities, and area differences are included in Table 3.


**Head-to-Head Comparison (DRE and PROSTest)**


A direct comparison of DRE and PROSTest performance across PSA strata ≥ 3 ng/mL revealed that PROSTest significantly outperformed DRE on all diagnostic metrics (Figure 3).

PROSTest demonstrated consistently higher sensitivity, specificity positive and negative predictive values, and overall diagnostic accuracy. All comparative differences between PROSTest and DRE reached statistical significance (*p* < 0.00001), underscoring the superior diagnostic utility of PROSTest in risk stratification and biopsy decision-making. The 3-point ROC analysis (Figure 4) confirmed the utility of the PROSTest in both PSA cohorts evaluated, the significant superiority of PROSTest, and its independence of DRE. The AUCs for PROSTest in the whole cohort and the two populations with PSA of 3–10 ng/L and >10 ng/mL were 0.9–0.92, while these were 0.52–0.57 for DRE.

### 3.3. Multivariate Analysis: PROSTest as the Principal Predictor of Prostate Cancer

To evaluate the independent contributions of clinical variables to PCa detection, a multivariate logistic regression model was constructed, incorporating age > 75 years, DRE status, family history of PCa, PSA > 10 ng/mL, and PROSTest result as covariates (Table 4). Ethnicity was not included because of so few PCa cases in the African American cohort. Among the five predictors, PROSTest had the highest OR (OR: 154.0) and was significantly (*p* < 0.0001) associated with a PCa diagnosis. In contrast, ORs ranged from 0.30 (family history) to 2.26 (Age > 75 years).

## 4. Discussion

Prostate cancer remains a significant global health concern, representing the most common non-skin cancer diagnosed among subjects and a leading cause of cancer-related mortality worldwide [1]. Although PSA testing has revolutionized PCa detection since its introduction, it lacks cancer specificity and is frequently elevated in benign conditions such as BPH and prostatitis [5]. Digital rectal examination, once the cornerstone of early detection, continues to be recommended in combination with PSA in many clinical guidelines despite long-standing concerns about its diagnostic limitations [12,13].

Recent evidence from a comprehensive meta-analysis by Matsukawa et al. (2024) underscores the limited diagnostic value of DRE, both as a standalone tool and in combination with PSA [30]. The pooled PPV of DRE across eight prospective studies was just 0.21 (95% CI: 0.13–0.33), with a cancer detection rate (CDR) of only 0.01 (95% CI: 0.01–0.02). These values were nearly identical to those of PSA alone in terms of PPV (0.22), but PSA demonstrated a significantly higher CDR (0.03; *p* < 0.01). Critically, the combination of DRE and PSA did not significantly outperform PSA alone on either metric (PPV: 0.19; CDR: 0.03), raising further questions about the clinical value of maintaining DRE as part of early detection strategies. A second meta-analysis by Naji *et al*. identified pooled sensitivities, specificities, PPV, and NPV of 51%, 59%, 41%, and 64%, respectively [31]. This further underscores the limited utility of DRE in stratification protocols.

In this study, we evaluated the clinical utility of DRE when used alongside PROSTest, a blood-based, multigene expression assay designed to guide biopsy decisions in men with elevated PSA. Our findings strongly suggest that PROSTest provides robust and independent risk stratification across PSA levels, and that DRE offers no additional diagnostic value in this context.

Consistent with the prior literature and the Matsukawa meta-analysis [30], DRE demonstrated limited diagnostic performance. In our own cohort, DRE achieved an accuracy of only 55.8–56.3% and sensitivity ranging from 42.0 to 64.4%, regardless of PSA level. These real-world findings echo the systematic review’s conclusion that DRE contributes minimally to cancer detection, and its added value over PSA is negligible in asymptomatic, screen-detected populations [30].

A quartile evaluation of PROSTest results identified that the clinically validated cut-off of ≥50 provided the most accurate value for analysis, although higher scores, e.g., >60, may be useful in individuals with PSA > 10 ng/mL. Using the ≥50 cut-off, PROSTest demonstrated significantly higher diagnostic performances than DRE and PSA. Among subjects with PSA 3–10 ng/mL, PROSTest achieved a sensitivity of 96.6% and an overall accuracy of up to 94.1%. These results were replicated in the higher PSA cohort (>10 ng/mL), where PROSTest again outperformed DRE across all metrics, irrespective of ethnicity or a family history of disease. Notably, PROSTest was effective in identifying cancers even when DRE was negative, and conversely, a positive DRE in the context of a negative PROSTest carried a negligible likelihood of malignancy (PPV as low as 1.9%).

Multivariate analysis further reinforced these findings. When PROSTest status, PSA > 10 ng/mL, age, family history, and DRE results were entered into a logistic regression model, PROSTest emerged as the strongest independent predictor of biopsy-confirmed prostate cancer (OR: 154; *p* < 0.0001). Neither PSA level > 10 nor DRE status reached statistical significance. These results align with prior reports showing that molecular profiling outperforms traditional biomarkers in early detection [17,21]. These results underscore that PROSTest is a robust, independent predictor of PCa risk, outperforming both PSA strata and DRE or their combination in multivariate analysis. The findings support the use of PROSTest as a standalone decision-making tool for biopsy guidance, independent of DRE findings or PSA levels.

A direct head-to-head comparison of DRE and PROSTest confirmed the superiority of PROSTest. Statistically significant differences were observed across all diagnostic categories (sensitivity, specificity, PPV, NPV, and accuracy), with PROSTest outperforming DRE in both PSA strata (*p* < 0.00001 for all comparisons). These differences were retained irrespective of ethnicity or family history of prostate cancer.

In light of the findings by Matsukawa [30] and our own cohort results, it is reasonable to conclude that DRE no longer offers additive value in risk stratification when molecular assays like PROSTest are available. Beyond its diagnostic limitations, DRE is known to be invasive, uncomfortable for subjects, and highly operator-dependent [10,15], factors that may further limit its utility in modern clinical workflows.

It is important to highlight some limitations of this study. It is retrospective (potential selection bias) and was statistically powered for a prevalence of 40% PCa. The prevalence of PCa in screening populations may be lower. The low number of African Americans with a PCa precluded a formal evaluation of ethnicity in the logistics model. All samples were evaluated at a single facility. However, this is a real-world study and the results have clear health policy implications. From a payer and guideline perspective, the data fulfills evidentiary standards for test utility and supports the removal of DRE from routine risk assessment when PROSTest is used. This supports the inclusion of molecular testing in streamlined, evidence-based algorithms focused on precision diagnostics.

## 5. Conclusions

PROSTest demonstrated high diagnostic accuracy and reproducibility in risk stratification for PCa among men with PSA ≥ 3 ng/mL, independent of DRE findings or PSA levels. In multivariate analysis, PROSTest was the only variable independently associated with PCa diagnosis. These findings may inform future revisions of clinical protocols regarding the role of PROSTest in this context to minimize biopsies falsely alerted by PSA or DRE findings.

## Figures and Tables

**Figure 1 cancers-17-03908-f001:**
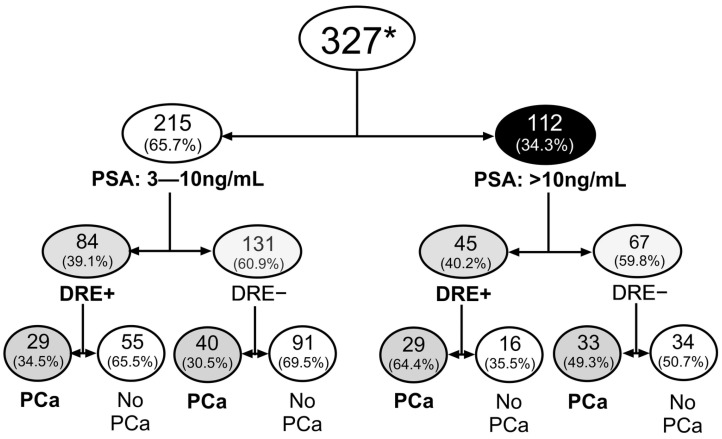
Distribution of subjects and biopsy outcomes stratified by PSA groups (3–10 ng/mL and >10 ng/mL). There was no significant difference in the proportion of DRE-positive (DRE+) cases between the two PSA strata (χ^2^ = 0.038, *p* = 0.85). However, the frequency of prostate cancer diagnoses was significantly higher in the elevated PSA group (>10 ng/mL) compared to the lower PSA group (χ^2^ = 16.54, *p* < 0.0001). * Of the 327 subjects, only 4 were above the age of 75 years, of which one had PSA of 3.7 ng/mL and the remaining three had PSA of 6.4–14.9.

**Figure 2 cancers-17-03908-f002:**
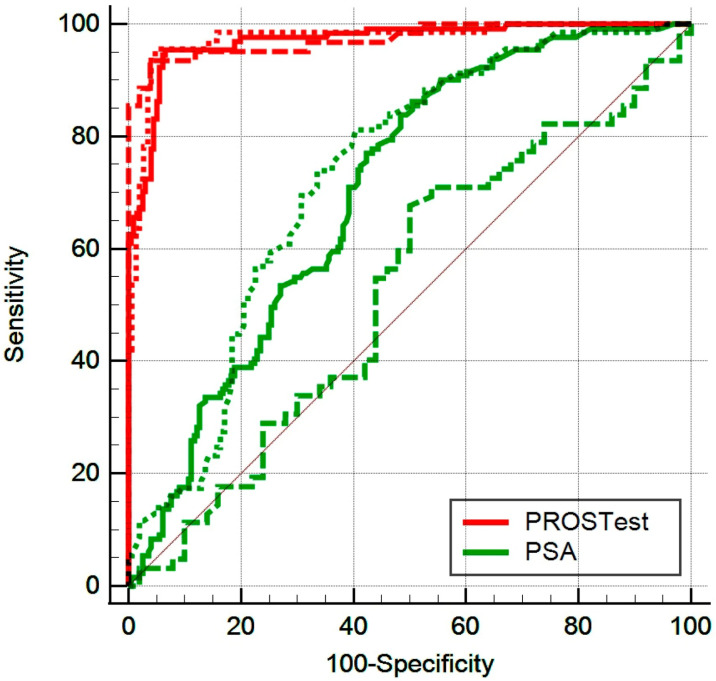
Head-to-head comparison of PSA and PROSTest for PCa detection. PROSTest demonstrated significantly higher AUROCs (*p* < 0.0001) than PSA across the whole cohort and both PSA strata. Solid line (all patients n 327), dotted lines (Cohort 1: *n* = 215), dashed lines (Cohort 2: *n* = 112).

**Figure 3 cancers-17-03908-f003:**
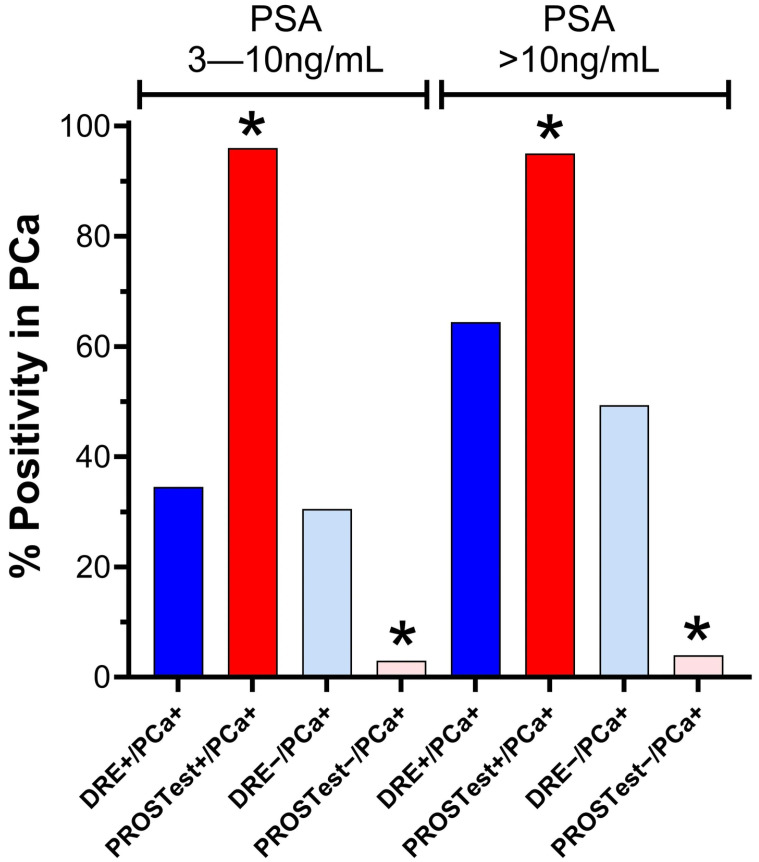
Head-to-head comparison of DRE status and PROSTest for PCa detection stratified by PSA levels (3–10 ng/mL and >10 ng/mL). PROSTest demonstrated significantly higher diagnostic accuracy than DRE across both PSA strata. Bars represent the percentage of subjects with biopsy-confirmed PCa in each DRE/PROSTest subgroup. * *p* < 0.00001 vs. corresponding DRE group (Chi-square test). Abbreviations: DRE, digital rectal examination; PCa, prostate cancer.

**Figure 4 cancers-17-03908-f004:**
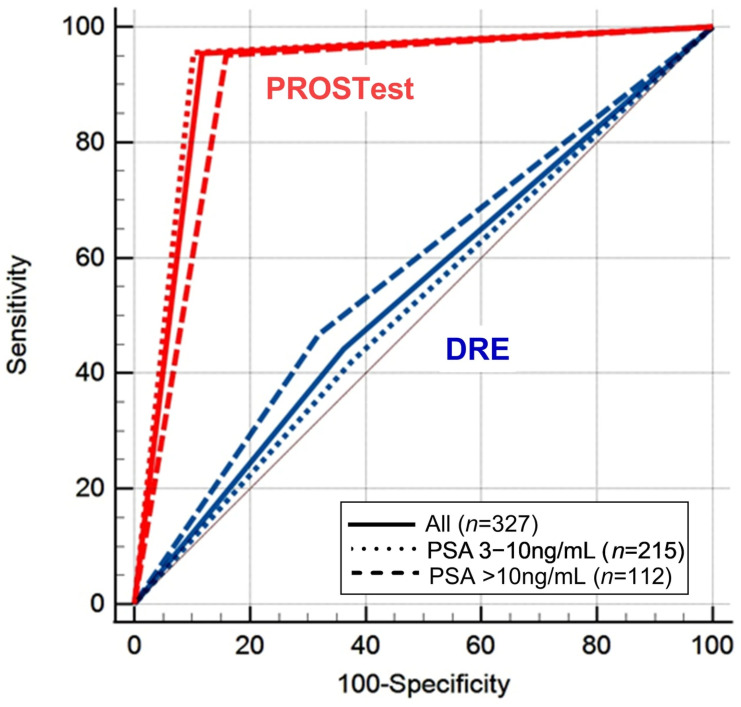
Head-to-head comparison of DRE status and PROSTest for PCa detection stratified by PSA levels (3–10 ng/mL and >10 ng/mL). PROSTest demonstrated significantly higher AUROCs (*p* < 0.0001) than DRE across the whole cohort and both PSA strata. Abbreviations: DRE, digital rectal examination.

**Table 1 cancers-17-03908-t001:** Odds ratio evaluation of PROSTest quartiles * for detecting prostate cancer.

Variable	OR Variable (95% CI)	Z-Statistic	*p*-Value Variable
Q1 (1.21–9.88)	0.011 (0.002–0.08)	4.454	<0.0001
Q2 (9.89–39.40)	0.036 (0.011–0.115)	5.547	<0.0001
Q3 (39.41–85.55)	2.404 (1.451–3.986)	3.402	<0.0001
Q4 (85.56–99.20)	614.3 (37.46–10,075.1)	3.402	<0.0001
Q3 + Q4 (39.41–99.20)	136.44 (47.38–392.89)	9.110	<0.0001
PROSTest + ve (≥50)	156.7 (61.98–396.17)	10.681	<0.0001

* Analysis versus median quartile. Abbreviations: CI, confidence interval; PROSTest+, positive PROSTest result. Note: PROSTest (cut-off 50) was the one variable that outperformed a median PROSTest (Q3 + Q4) result, indicating a cut-off of 50 was appropriate.

**Table 2 cancers-17-03908-t002:** Diagnostic performance of PROSTest in DRE-positive and DRE-negative subjects, stratified by PSA cohort (3–10 ng/mL and >10 ng/mL).

	PSA 3–10 ng/mL				PSA > 10 ng/mL			
	DRE+/ PROS+	DRE+ /PROS−	DRE−/ PROS+	DRE−/ PROS−	DRE+/ PROS+	DRE+ /PROS−	DRE−/ PROS+	DRE−/ PROS−
Number	32	52	49	82	30	15	37	30
PCa	28	1	38	2	28	1	31	2
Non-PCa	4	51	11	80	2	14	6	28
Sensitivity	96.6% (82.2–99.9%)		95% (83.1–99.4%)		96.6% (82.2–99.9%)		93.9% (79.8–99.3%)	
Specificity	92.7% (82.4–98.0%)		87.9% (79.4–93.8%)		87.5% (61.6–98.5%)		82.4% (65.5–93.2%)	
PPV	87.5% (73.1–94.8%)		77.6% (66.4–85.8%)		93.3% (79.3–98.1%)		83.8% (71.3–91.5%)	
NPV	98.1% (88.1–99.7%)		97.6% (91.2–99.4%)		93.3% (66.9–99.0%)		93.3% (78.4–98.2%)	
Accuracy	94.1% (86.7–98.0%)		90.1% (83.6–94.6%)		93.3% (81.7–98.6%)		88.1% (77.8–94.7%)	

Abbreviations: DRE, digital rectal examination; PROS, PROSTest; PCa, prostate cancer; PPV, positive predictive value; NPV, negative predictive value. Note: Each cell represents a subgroup defined by DRE and PROSTest results within the indicated PSA range.

**Table 3 cancers-17-03908-t003:** Metrics comparing PSA with PROSTest for detecting prostate cancer.

Cohort	AUC (PSA)	Sens/Spec	Cut-Off	AUC (PROS)	Sens/Spec	Cut-Off	Difference in AUC	*p*-Value
1 (*n* = 215)	0.73 ± 0.035	81.2%/59.6%	>5.5 ng/mL	0.97 ± 0.011	95.7%/95.2%	>60.06	0.245 ± 0.04	<0.0001
2 (*n* = 112)	0.53 ± 0.056	67.7%/50.0%	>14.9 ng/mL	0.97 ± 0.013	93.6%/96.0%	>82.4	0.446 ± 0.06	<0.0001
All PSA (*n* = 327)	0.70 ± 0.028	86.3%/49.5%	>5.9 ng/mL	0.91 ± 0.001	95.4%/93.4%	>60.06	0.268 ± 0.03	<0.0001

Abbreviations: AUC, area under the curve; PROS, PROSTest; Sens, sensitivity, Spec, specificity. Note: PROSTest outperformed PSA for detecting prostate cancers.

**Table 4 cancers-17-03908-t004:** Multivariate logistic regression model assessing independent predictors of prostate cancer.

Independent Variables	OR	95% CI	Co- Efficient ±SE	Wald	*p*-Value
Age > 75 years	2.26	0.486 to 10.51	0.851 ± 0.78	1.08	0.30
DRE + ve	1.48	0.625 to 3.50	0.392 ± 0.44	0.79	0.37
Family History	0.30	0.092 to 0.965	−1.209 ± 0.60	4.08	0.04
PSA > 10 ng/mL	1.78	0.765 to 4.144	0.577 ± 0.43	1.79	0.18
PROSTest + ve	154.0	59.5 to 398.9	5.037 ± 0.48	107.74	<0.0001
Constant	-	-	−3.616 ± 0.49	54.22	<0.0001

Abbreviations: SE, standard error; CI, confidence interval; DRE+, positive digital rectal examination; PROSTest+, positive PROSTest result. Note: PROSTest was the one variable to reach statistical significance in the model (*p* < 0.0001), with the highest OR and co-efficients, indicating it is the principal predictor of prostate cancer in this analysis.

## Data Availability

Due to privacy and ethical concerns, the data that support the findings of this study are not publicly available but are available on request from the corresponding author.

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
