# Peer review of "PROSTest, a Novel Liquid Biopsy Molecular Assay, Accurately Guides Prostate Cancer Biopsy Decision-Making in Men with Elevated PSA Irrespective of DRE Findings"

_cancers, 2025, doi:10.3390/cancers17243908_

Round 1

Reviewer 1 Report

Comments and Suggestions for Authors

The Manuscript is about a novel liquid biopsy molecular assay in patients in PCa screening. The subject is interesting and the Manuscript is well-written, but some points should be highlighted:

  1. A patient selection flowchart and its description should be added;
  2. How many urologists did the DREs? How expert were these urologists?
  3. Limitations of the study should be added, such as the single lab analyses;
  4. Why did the Authors not compare PROSTest with MRI with or without DRE?

Author Response

Comments to the Authors:

The Manuscript is about a novel liquid biopsy molecular assay in patients in PCa screening. The subject is interesting and the Manuscript is well-written, but some points should be highlighted.

Response: We thank the reviewer for the time taken to review the paper and for the insights provided.

Comments 1. A patient selection flowchart and its description should be added.

Response 1. The study included all patients prospectively evaluated for prostate biopsy between January 2022-June 2024. The inclusion criteria included an age ≥45 years and a PSA ≥3ng/mL. This information is included in the text Materials and Methods (lines 104-109) and in the Supplement as requested (Supplemental Figure S1).

Comments 2. How many urologists did the DREs? How expert were these urologists?

Response 2. Two board-certified urologists undertook the DRE evaluations (line 122).

Comments 3. Limitations of the study should be added, such as the single lab analyses.

Response 3. We have included this in the Discussion (lines 377-380).

Comments 4. Why did the Authors not compare PROSTest with MRI with or without DRE?

Response 4. We thank the reviewer for raising this question. The goal of this study was to compare PROSTest with DRE. The question of MRI is an important one and is the subject of a separate prospective study that has been submitted for peer-review.

Reviewer 2 Report

Comments and Suggestions for Authors

In the paper the authors used the data from PROSTest, a novel liquid biopsy molecular assay, to determine the test characteristics and improvement in prostate cancer (PCa) detection in men with various levels of PSA and DRE findings. The study is very interesting and novel as the PROSTest can be very useful in improvement of detection of PCa particularly in high-risk men. 

However, there are some points that the authors need to address to better understand the implication of their findings. 

1. It would be helpful if the authors provide some information about demographic (e.g. average age, race/ethnicity) and clinical characteristics for all men included in the study. Since race/ethnicity and family history (FH) of PCa are strong risk factors it would be helpful to report on the proportion of men who are African American and those with a FH of PrCa. 

2. In the introduction the authors need  to provide additional background information for PROSTest. When it was introduced, and have other studies provided any data on PPV and NPV for this test?   

3. The methods need to describe a bit more in detail the PROSTest assay (maybe add a supplemental table with the list of all genes) and how the scoring system was developed.

4. It is not clear why the authors use a PSA>3 ng/ml as cut-off for this study. In the US the current guidelines use PSA > 4ng/ml so it would help to clarify this point. Also why men were categorized in two groups: PSA 3-10 vs PSA>10? This needs to be clarified.  

5. Table 3: Although I understand that the authors used PROSTest value of 50 to define high likelihood of positivity, it might be useful to categorize the test in quartiles using 25, 50 and 75 cutoff points for multivariate logistic regression model and compare it's performance with using the median cutoff. Also since the authors are running logistic regression models please show ORs and 95% CI for the variables. 

6. Figure 3 needs to show AUC values also using a continuous PROSTest values (or at least use quartiles) and continuous PSA values. The authors can compare the AUCs between the current figure and the ones using continuous values. That might help to pick better cutoff points. 

7. Did the authors evaluate the PPV, NPV and accuracy for high-risk men (e.g. African American or men with FH of PrCa) as well as men with younger ages at diagnosis? These subgroup analyses might be useful.  

8. The discussion is missing the limitations of the study. The authors need to include that information. Also what statistical power did the study have with only 327 men? This information needs to be added.   

Author Response

Overall comments:

In the paper the authors used the data from PROSTest, a novel liquid biopsy molecular assay, to determine the test characteristics and improvement in prostate cancer (PCa) detection in men with various levels of PSA and DRE findings. The study is very interesting and novel as the PROSTest can be very useful in improvement of detection of PCa particularly in high-risk men. However, there are some points that the authors need to address to better understand the implication of their findings.

Response: We thank the reviewer for the positive comments about the study and appreciate the opportunity to address the important suggestions made for improvements.

Comments 1. It would be helpful if the authors provide some information about demographic (e.g. average age, race/ethnicity) and clinical characteristics for all men included in the study. Since race/ethnicity and family history (FH) of PCa are strong risk factors it would be helpful to report on the proportion of men who are African American and those with a FH of PrCa.

Response 1. Thank you for raising this. We have examined patient records for ethnicity and a PCa family history and have included this information (lines 173-175).

Comments 2. In the introduction the authors need to provide additional background information for PROSTest. When it was introduced, and have other studies provided any data on PPV and NPV for this test?

Response 2. We have updated the introduction appropriately. The test was introduced in 2024 (line 83) and two recent studies (Prostate 2025, published online Sept 19 and Oct 23) have identified diagnostic metrics (PPV and NPV) (lines 90-93, new references 22 and 23).

Comments 3. The methods need to describe a bit more in detail the PROSTest assay (maybe add a supplemental table with the list of all genes) and how the scoring system was developed.

Response 3. Thank you for the suggestion. We have amended this section to reflect this and have added a supplemental table (Supplemental Table S1) as requested by the reviewer (lines 148-152).

Comments 4. It is not clear why the authors use a PSA>3 ng/ml as cut-off for this study. In the US the current guidelines use PSA > 4ng/ml so it would help to clarify this point. Also why men were categorized in two groups: PSA 3-10 vs PSA>10? This needs to be clarified.

Response 4. Thank you for raising this question. We chose 3ng/mL as a lower limit because values between 3-4ng/mL are typically considered within a borderline to intermediate range that may require additional testing. This has been recently recognized by the NCCN. The risk of prostate cancer in men with these values ranges between 15-27% identifying this as an important sector to include in an analysis. Values >10ng/mL are considered significantly elevated and strongly associated with a high risk (>50%) of clinically significant prostate cancer. Evaluated these two cohorts would allow us to define utility of the test in an appropriate risk population (3-10ng/mL) as well as in a high-risk population (>10ng/mL). This is included (lines 215 and 241).

Comments 5. Table 3: Although I understand that the authors used PROSTest value of 50 to define high likelihood of positivity, it might be useful to categorize the test in quartiles using 25, 50 and 75 cutoff points for multivariate logistic regression model and compare it's performance with using the median cutoff. Also since the authors are running logistic regression models please show ORs and 95% CI for the variables.

Response 5. Thank you for suggesting this approach. We have undertaken the analyses as requested and included the data (Section 3.2.1, Table 1). Furthermore, we have included the ORs and 95% CIs in the regression model table (Table 4).

Comments 6. Figure 3 needs to show AUC values also using a continuous PROSTest values (or at least use quartiles) and continuous PSA values. The authors can compare the AUCs between the current figure and the ones using continuous values. That might help to pick better cutoff points.

Response 6. We have included a head-to-head comparison (lines 260-275) that evaluates PROSTest versus PSA using continuous data. A figure is provided (new Figure 2) comparing AUROCs for both biomarkers in the whole cohort (n=327) and in each of the two sub-cohorts (Cohort 1: PSA 3-10ng/mL, n=215; Cohort 2: PSA >10ng/mL, n=112). Table 3 (new) includes the metrics for comparisons as well as the different cut-offs derived from the AUCs.

Comments 7. Did the authors evaluate the PPV, NPV and accuracy for high-risk men (e.g. African American or men with FH of PrCa) as well as men with younger ages at diagnosis? These subgroup analyses might be useful.

Comments 7. Thank you for the suggestion. The sub-analyses was performed and identified no impact of ethnicity, a family history or age on the PPV, NPV and accuracy of PROSTest in DRE+ and DRE- individuals (lines 230-239 and lines 250-259, respectively).

Comments 8. The discussion is missing the limitations of the study. The authors need to include that information. Also what statistical power did the study have with only 327 men? This information needs to be added.

Comments 8. Thank you for identifying this. We have updated the discussion to reflect limitations as requested (lines 377-380). A formal power-analysis was undertaken for this study; we apologize for not originally including this. This is now in the Methods (lines 109-117).

Round 2

Reviewer 1 Report

Comments and Suggestions for Authors

The Authors followed Reviewers' suggestions and the Manuscript is now suitable for publication

Author Response

Comments 1: The Authors followed Reviewers' suggestions and the Manuscript is now suitable for publication

Response 1: We thank the reviewer for their kind comments.

Reviewer 2 Report

Comments and Suggestions for Authors

The authors have addressed most of the reviewers' comments and the manuscript has improved. However, there are still some methodological issues that need to be corrected. 

  1. Methods: Assumptions about sample size calculations (lines 110-116). Assuming a prostate cancer prevalence of 40% is probably not correct unless this is a very high-risk population. What are the required sample size for prevalence estimates of 10%, 20% and 30%... these need to be presented or discussed in the Discussion section.
  2.  The authors need to present patients' characteristics in a Table (either main or supplement) and compare characteristics and PSA values for men with and without PCa. 
  3. Were the PSA values corrected for prostate volume or BMI? Men with large PSA have lower circulating values of PSA due to hemodilution effect. 
  4. Table 1 seems wrong. How were the cutoff points for PROTest determined? Were they based on all men or men without prostate cancer? We usually compare highest to lowest quartiles of distribution with Q1 serving as reference category. The OR fluctuations from one quartile to another indicate non-linear relationship.... something is off here. 
  5. Please present results of sub-analyses (Lines 228-237 and 248-257) as supplementary tables. 

Author Response

Comments and Suggestions for Authors

The authors have addressed most of the reviewers' comments and the manuscript has improved. However, there are still some methodological issues that need to be corrected.

We thank the reviewer for the time taken to review the resubmission as well as the comments raised.

Comments 1: Methods: Assumptions about sample size calculations (lines 110-116). Assuming a prostate cancer prevalence of 40% is probably not correct unless this is a very high-risk population. What are the required sample size for prevalence estimates of 10%, 20% and 30%... these need to be presented or discussed in the Discussion section.

Response 1: This is a good question as it raises test accuracy under different conditions. We, however, respectfully disagree with the statement that the power calculation may be incorrect. Men who see their urologist with an elevated PSA, with an abnormal DRE and/or family history of prostate cancer, per definition, are at an increased risk of the disease. In this study, the disease prevalence is 40.1% (131/327). Published articles (Chen et al., 2023, and Lophatananon et al, 2023) identified prostate cancer prevalence rates of 38.7% – 61.3% (weighted average of 40.7%) in subjects with PSA ≥3ng/mL. We acknowledge that in a general screening study (all comers) the prevalence could be lower. We have accordingly raised this in the discussion (lines 374-375).

 Chen Y XD, Ruan M, et al. A prospective study of the prostate health index density and multiparametric magnetic resonance imaging in diagnosing clinically significant prostate cancer. Investig Clin Urol 2023; 64: 363–72.

Lophatananon A LA, Burns-Cox N, et al. Re-evaluating the diagnostic efficacy of PSA as a referral tes to detect clinically significant prostate cancer in contemporary MRI-based image-guided biopsy pathways. . J Clin Urol 2023; 16: 264–73.

Comments 2: The authors need to present patients' characteristics in a Table (either main or supplement) and compare characteristics and PSA values for men with and without PCa.

Response 2: Thank you for suggesting this. We have included a Supplementary Table 2 with the information, as requested. This also includes a statistical comparison in PSA values for men with and without PCa

Comments 3: Were the PSA values corrected for prostate volume or BMI? Men with large PSA have lower circulating values of PSA due to hemodilution effect.

Response 3: Thank you for the question. No, PSA were not corrected based on prostate volume or BMI. The data included in the analysis are standard measurements. We include this in the methods (line 142).

Comments 4: Table 1 seems wrong. How were the cutoff points for PROTest determined? Were they based on all men or men without prostate cancer? We usually compare highest to lowest quartiles of distribution with Q1 serving as reference category. The OR fluctuations from one quartile to another indicate non-linear relationship.... something is off here.

Response 4: Thank you – we appreciate you identifying this. The OR have been recalculated for the four quartiles for PROSTest (based on all men). The ORs, as expected, increase with the 4th quartile exhibiting the highest ratio. As a comparator we also examined the OR for Q3+Q4 vs. Q1+Q2 – as this would reflect a cut-off for the assay of 39.41. This was 136.4 (z-statistic: 9.11). The OR for the PROSTest (using the clinically validated cut-off of 50) had an OR of 156.7 (z-statistic: 10.68). This information is included in the updated table (Main document – Table 1).

Comments 5: Please present results of sub-analyses (Lines 228-237 and 248-257) as supplementary tables. 

Response 5: We have included this information in the Supplement (Supplementary Figure S2A-C). This includes the 2x2 tables as well as the diagnostic outputs.